# Exploring the Function of Quorum Sensing Regulated Biofilms in Biological Wastewater Treatment: A Review

**DOI:** 10.3390/ijms23179751

**Published:** 2022-08-28

**Authors:** Sania Sahreen, Hamid Mukhtar, Kálmán Imre, Adriana Morar, Viorel Herman, Sundas Sharif

**Affiliations:** 1Institute of Industrial Biotechnology, Government College University, Lahore 54000, Pakistan; 2Department of Animal Production and Veterinary Public Health, Faculty of Veterinary Medicine, Banat’s University of Agricultural Sciences and Veterinary Medicine “King Michael I of Romania”, 300645 Timisoara, Romania; 3Department of Infectious Diseases and Preventive Medicine, Faculty of Veterinary Medicine, Banat’s University of Agricultural Sciences and Veterinary Medicine “King Michael I of Romania”, 300645 Timisoara, Romania

**Keywords:** biofilm, quorum sensing, wastewater treatment, quorum quenching, acyl-homoserine lactone, autoinducers, QS regulation

## Abstract

Quorum sensing (QS), a type of bacterial cell–cell communication, produces autoinducers which help in biofilm formation in response to cell population density. In this review, biofilm formation, the role of QS in biofilm formation and development with reference to biological wastewater treatment are discussed. Autoinducers, for example, acyl-homoserine lactones (AHLs), auto-inducing oligo-peptides (AIPs) and autoinducer 2, present in both Gram-negative and Gram-positive bacteria, with their mechanism, are also explained. Over the years, wastewater treatment (WWT) by QS-regulated biofilms and their optimization for WWT have gained much attention. This article gives a comprehensive review of QS regulation methods, QS enrichment methods and QS inhibition methods in biological waste treatment systems. Typical QS enrichment methods comprise adding QS molecules, adding QS accelerants and cultivating QS bacteria, while typical QS inhibition methods consist of additions of quorum quenching (QQ) bacteria, QS-degrading enzymes, QS-degrading oxidants, and QS inhibitors. Potential applications of QS regulated biofilms for WWT have also been summarized. At last, the knowledge gaps present in current researches are analyzed, and future study requirements are proposed.

## 1. Introduction

Since the start of microbiology, microorganisms have always been considered as freely suspended cells, having specific physiological, morphological, and growth characteristics in the culture media. However, microbes have been found to interact closely with interfaces and surfaces to form aggregates. These microbial aggregates are usually glued together with a sticky secretion of slime [1,2]. Such types of bacterial associations where a cluster of microbial cells are attached to a surface are termed biofilms (a primary mode of living and growth for bacteria) [3]. The initiation of biofilm formation requires a surface and the availability of nutrients vital for the growth of bacteria [4,5]. Costerton et al. (1978) defined biofilms as an aggregation of single or diverse types of microbial cells, attached to a biotic or abiotic surface, enclosed in a covering of extrapolymeric substance (EPS), where they live in a cooperative manner [6]. In natural reservoirs, various bacterial species, algae, protozoa and fungi form biofilms. About 50% of the total thickness of biofilm (0.2–1.0%) is formed by EPS, and the rest of the portion contains microorganisms [7]. Besides the aqueous matrix and microbial cells, biofilm also contains complex secreted metabolite polymers, cell lysis products, absorbed nutrients, and particulate material [8]. Hence, macromolecules, such as proteins, lipids, polysaccharides, and even DNA and RNA, have also been observed in the biofilm environment [3]. Biofilms also contain ketal-linked pyruvates and uronic acids, i.e., mannuronic acids, D-glucuronic acid, and D-galacturonic acid, which characterize the anionic nature of the biofilm [8]. These particles diffuse in and out of the biofilm, on the basis of their water-binding affinity, mobility, and size [8]. The biofilm structure also influences the mass transport of the available substances. The transportation of solutes is driven via pores and water channels in biofilms [9]. Due to the presence of numerous micro-environments, the biofilm matrix exhibits a high degree of microheterogeneity [5].

Biofilm formation is a dynamic process, whose development has several steps, including the reversible adherence of planktonic bacteria to the surface via van der Waals interaction, irreversible attachment through pili, fimbriae, and flagella, bacterial proliferation and secretion of EPS, followed by quorum sensing (QS) and biofilm maturation (Figure 1) [10,11,12].

Biofilms protect microbial cells from adverse environmental conditions, i.e., pH changes, presence of free radicals or toxic substances and less availability of nutrients as well as antibacterial agents and toxic actions [13]. Within a biofilm, the microbial population ranges from 10^8^ to 10^11^ microbial cells per g wet weight [14]. The protective EPS layer of biofilm also prevents the microbes from exposure to pesticides, heavy metals, and hydrocarbons [15]. EPS surfactants are capable of solubilizing organic pollutants, which are usually non-degradable [16,17,18]. This advanced tolerance property of biofilm has specific relevance to biotransformation and bioremediation [19,20]. The composition of biofilm is extremely complex, and several techniques are being used to determine its morphology and composition. Atomic force microscopy (AFM), scanning electron microscope (SEM), and confocal laser scanning microscopy (CLSM) are used to determine the roughness, stiffness, and topography of mature biofilms. SEM and Fourier transform infrared (FTIR) are also used to determine the structure of EPS and the interactions among bacterial species in the biofilm. The elements present in the biofilm are also analyzed by energy dispersive X-ray (EDX) spectroscopy, while surface plasmon resonance (SPR) spectroscopy can analyze the whole biofilm formation process. The biofilm growth and microbial adhesion to surfaces are determined by electrical impedance spectroscopy (EIS). The bacterial diversity in a biofilm is assessed by 16S rRNA sequence analysis, and crystal violet assay is used to estimate the growth of biofilm (Figure 2) [21,22].

## 2. Relationship between Quorum Sensing (QS) and Biofilm Formation

QS is a type of bacterial cell–cell communication, where bacteria produce and release chemical molecules to regulate their gene expressions, responding to the cell population density [23,24,25]. This process was first proposed when bacterial interactions in *Streptococcus pneumoniae* and *Vibrio fischeri* were described by Miller and Nealson in the 1970s [24,26,27,28]. In 1998, QS influence in biofilm formation was described in *Pseudomonas aeruginosa.* However, later studies have revealed that the effect of QS on biofilm formation in *P. aeruginosa* was due to unnatural experimental conditions [29]. *P. aeruginosa* exhibits several QS circuits, which affect the biofilm development. At the genetic level of *P. aerogenosa*, both las and rhl acyl-HSL QS systems show involvement in the production of rhamnolipids required to maintain the architecture and structure of biofilm; hence, understanding the association between QS and the formation of biofilm is a complicated matter [4,30]. In QS, bacteria produce autoinducers (AIs), small signaling molecules, and release them into their surroundings. The concentrations of AIs are proportional to the bacterial population [31]. With the propagation of bacteria, the bacterial population increases, and hence, the concentrations of autoinducers also increase [4]. When a concentration of AIs attains a certain threshold level, they bind with cognate receptors. This binding triggers the downstream gene expression, which controls several bacterial activities, i.e., biofilm formation, virulence factor secretion, bioluminescence, sporulation, antibiotic production, etc. [32,33]. In this way, QS enables the bacteria to survive in a continuously changing environment [34]. In mature biofilms, QS also controls the changes required to allow the entry of nutrients into the cells. These changes usually include the formation of pores, channels, and pillar-like structures. This type of architecture ensures adequate division of nutrients among the cell population in a high-density biofilm. QS also influences biofilm development, as it has an important role in bacterial accumulation on solid surfaces [34].

## 3. QS Signaling Molecules

Bacteria produce and respond to a diverse variety of AIs; however, there are three well-defined classes of autoinducers that are predominately produced by bacteria. These include acyl homoserine lactones (AHLs), autoinducer oligopeptide (AIP), and autoinducer-2 (AI-2). Usually, Gram-negative bacteria produce AHLs, and Gram-positive bacteria produce AIPs, while AI-2 is used by both types of bacteria for interspecies communication [24,33,35]. Besides these major signaling molecules, streptomyces butyrolactones, indole and diketopiperazines (DKPs) also act as QS molecules in some bacteria [36,37,38,39]. These diverse chemical molecules provide a diverse platform of intraspecies and interspecies crosstalk [4]. Among all these molecules, AHLs are widely studied and characterized as bacterial interspecies signaling molecules [40]. Some bacteria produce more than one type of signaling molecule, which makes the quorum sensing mechanism quite complex to understand [24,31].

### 3.1. Classical QS Signaling Molecules

#### 3.1.1. Acyl Homoserine Lactones

Acyl homoserine lactones (AHL) are specific to Gram-negative bacteria for their cell-to-cell communication. These autoinducers consist of an acyl side chain attached to a homoserine lactone moiety by an amide linkage [41]. LuxR/I-type is the most studied quorum sensing system. In AHL-mediated communication, AHLs bind to the transcriptional factors, which initiate the signaling cascade to regulate the relevant genes. LuxI homologue synthesizes AHL synthase, which produces AHL molecules. AHL molecules, having short side-chains, easily diffuse across the membrane, while the molecules with larger side-chains used efflux pumps to come out of the cell. On reaching the high concentration, the AHL molecules are taken by the cells, and cytoplasmic LuxR proteins interact with them to regulate the downstream process [42]. Several Gram-negative bacteria, such as *P. aeruginosa, P. putida, Serratia liquefacians,* and *Burkholderia cepacia*, are cited in the literature to communicate through AHLs [43,44]. However, *Exiguobacterium* sp., a Gram-positive bacterium, also communicates with AHL signaling (Figure 3A) [45].

#### 3.1.2. Autoinducing Peptides

Autoinducing peptides (AIPs) are small post-transcriptionally modified peptides produced by Gram-positive bacteria [46,47]. These AIPs regulate the communication process by some specific surface receptors of the cells, via a chain of phosphorylation, and dephosphorylation series, usually by membrane-associated sensor kinases and response regulators of bacteria [44]. Studies report that Gram-positive bacteria, such as *S. aureus*, *Lactobacillus plantarum*, *Clostridium difficile*, *C. botulinum*, *C. perfringens*, *Enterococcus faecalis* and *Listeria monocytogenes*, communicate via AIPs (Figure 3B) [48,49,50]. 

#### 3.1.3. Autoinducer-2

Autoinducer-2 is produced via activated methyl cycle and is a derivative of 4, 5-dihydroxy-2, 3 pentanedione (DPD). AI-2 is used as a signaling molecules by both Gram-positive and Gram-negative bacteria. When the concentration of AI-2 reaches a threshold level, a signal transduction cascade is activated to uptake and process AI-2. Cell membrane transporters uptake AI-2 inside the cell where cell kinases carry out phosphorylation of AI-2 and activate the QS circuit by binding to the relevant activator and repressor genes. *Vibrio cholera*, *V. harveyi*, *S. typhi*, *Escherichia coli* and *Deinococcus radiodurans* are reported to use the AI-2 signaling pathway (Figure 3C) [51,52,53].

## 4. Role of QS-Regulated Biofilms in Wastewater Treatment (WWT)

Biological methods of wastewater treatment have long been adopted in wastewater bioremediation. The basic mechanism behind this technique is the utilization of microorganisms in contaminated environments, the contaminants of which act as the energy sources for the bacteria. Microorganisms uptake the contaminants, metabolize them and produce building blocks for their cells. As a result of this process, the contaminants are removed from the environment [4]. In several wastewater treatment systems, bacterial biofilms are employed to perform detrimental roles [54]. A dense microbial population is utilized in biological wastewater treatment in various forms, i.e., granules, flocs, and biofilms. Mostly Gram-negative bacteria producing AHLs are employed for wastewater treatment [54,55,56]. Wastewater with problematic biological oxygen demand (BOD), nitrogen, nitrate ammonia, and dissolved O2 is treated with bacterial biofilms. The organic and inorganic constituents of wastewater induce the growth of indigenous microbes, the metabolites of which also reduce water contamination [22]. The treatment of domestic wastewater and industrial wastewater is carried out in biological contact oxidation tanks, moving bed biofilm reactors (MBBRs), biological aerated filters, biological rotating discs, integrated fixed film activated sludge reactors, and biofilm fluidized beds (BFB). MBBRs, trickling filters, and granular sludge need robust biofilms for wastewater treatment [54]. These reactors eliminate organic and nitrogenous pollutants quickly and generate less sludge [21,57].

The biofilm use in wastewater bioremediation depends on the interaction of microbes with xenobiotic substances present in the environment. The microbial cells immobilized in the biofilm synthesize several enzymes, which contribute significantly to bioremediation. Biofilms are preferred over planktonic cells in performing efficient bioremediation because of the horizontal gene transfer among the members of a biofilm [25]. The functional regulation of biofilms is due to the process of QS. The study at the WWT facility found a strong correlation between AHL production and biofilm development at the growth or activation stage [4]. However, in later stages of biofilm formation, this connection was not statistically significant [58]. The QS signals, i.e., AHLs produced by *Pseudomonas, Agrobacterium*, and *Aeromonas*, significantly contribute to wastewater treatment [4]. AHLs also accelerate phenol degradation and its derivatives present in wastewater [59]. Hexadecane degradation is aided by the QS signals of *Acinetobacter* sp. DR1 [60]. The *P. aeruginosa* rhl QS system and catechol 2, 3-dioxygenase expression system are involved in benzoate, phenol, and phenanthrene degradation [61]. *Aeromonas* and *Pseudomonas*, the AHL-producing bacterial genera, are abundantly present in activated sludge and are also used for the purification of industrial and domestic wastewater. N-heptanoyl-L-homoserine lactone regulates the structure and metabolic activity of activated sludge in WWT [62]. Hexadecane is metabolized by the autoinducers produced by *Acinetobacter* sp. via QS. *Desulfovibrio* and other sulfate-reducing bacteria (SRB) are used in wastewater treatment containing Cu^+2^ [63]. *Acidovorax* sp., *Pseudomonas* sp., and *Luteimonas* sp. are used to accelerate the biodegradation of bisphenol A (BPA) in wastewater. The wastewater treatment systems for nitrogen metabolism have been reported to contain *Bosea*, *Devosia*
*Mezorhizobium*, *Paracocci,* and *Pseudoxanthomonas* species [64,65]. 

EPS synthesis, biofilm development, organic pollutant biodegradation, and biofouling control in wastewater treatment have all been reported to be influenced by quorum sensing (QS) and quorum quenching (QQ), a technique of blocking the QS system and inhibiting gene expression (Table 1) [22]. With the advancement in molecular biology and analytical detection techniques, the formation of AHLs and AHL-alike molecules was observed in a variety of biological wastewater treatment systems (Table 2) [66,67,68,69,70]. Moreover, it was found that many activated sludges included both AHL-producing QS bacteria and AHL-degrading QQ bacteria [59,66,71,72,73]. AHL signals, impressively, can play a significant role in the biological wastewater treatment process, particularly in the areas of biofilm formation and maturation [74,75,76,77,78], microbial aggregation and stabilization [59], exoenzyme activity [62], sludge structure stability and granule formation [79,80]. Additionally, the beneficial effects of AHL-based QS regulation in the biodegradation of organic pollutants, such as phenol biodegradation, ammonium oxidation [59,68,81] anthranilate degradation, and denitrification, have also been discovered [82].

## 5. QS-Mediated Biofilm Regulation Methods in Wastewater Treatment Systems (WWTs)

Once it was discovered that quorum sensing has an influence on biofilm formation in wastewater treatment systems, researchers shifted their focus to how to regulate QS in WWTs so as to improve these systems’ working efficiencies [88]. Studies were carried out in search of QS regulation methods to optimize wastewater treatment systems (Table 3) [113,114,115]. In an experiment aimed at aquaculture wastewater treatment with biofilm, QS was promoted in the system by the addition of two AHL signaling molecules i.e., N-hexanoyl-homoserine lactone (C6-HSL) and N-3-oxo-octanoyl-homoserine-lactone (3-oxo-C8-HSL) [116]. The addition of AHLs had a positive effect on biofilms; it not only significantly increased biofilm biomass, but also helped in improving the overall internal environment of the system. In another study on QS regulated biofilm, conducted by Valle et al. (2004), it was found that with the addition of 2 μmol/L AHLs in a methanol wastewater treatment system, not only was the methanol decomposition rate increased, but also a positive change in bacterial diversity, community composition, and community function in the system was observed [59].

All QS signaling molecules have different effects, and thus, QS-based regulation methods are categorized into two groups: QS enrichment methods and QS inhibition methods [117].

**Table 3 ijms-23-09751-t003:** QS regulatory systems and autoinducers present in both Gram-negative and Gram-positive bacteria.

Type of Microbe	Bacterial Specie	AI Molecule Type	QS Regulatory System	Ref.
Gram-negative bacteria [(LuxI/LuxR homologue) based regulator system]	*Erwinia carotovora*	N-(3-oxooctanoyl)-HSL	TraI/TraR	[118]
	*Serratia liquefaciens*	N-butanoyl-HSL	SwrI/SwrR	[119]
	*Yersinia pseudotuberculosis*	N-octanoyl-HSL, N-(3-oxohexanoyl)-HSL	YtbI/YtbR YpsI/YpsR	[120,121]
	*Burkholderia cepacia*	N-octanoyl-HSL	CepI/CepR	[24,122]
	*Halomonas anticariensis*	N-butanoyl-HSL, N-hexanoyl-HSL, N-octanoyl-HSL, N-dodecanoyl-HSL	hanR/hanI	[15,123]
Gram-positive bacteria	*Bacillus subtilis*	ComX, CSF (competence stimulating factor)	ComP/ComA	[6]
	*Staphylococcus intermedius*	AIP	AgrC/AgrA	[124]
	*Streptococcus mutans*	CSP ComC) ,XIP (sigx inducing peptide) ComS)	ComD/ComE ComR	[125,126]
	*Lactobacillus plantarum*	LamD558	LamC/LamA	[127,128]

### 5.1. QS Enrichment Methods

These methods simply help to increase the content of QS signaling molecules in the biofilms and thus help smoothen the overall operation of biological wastewater treatment systems [117]. There are three ways to enhance QS in WWTs: (i) addition of exogenous QS signaling molecules, (ii) increased synthesis of QS signaling molecules through addition of accelerators, and (iii) QS bacteria cultivation. 

#### 5.1.1. Addition of Exogenous QS Signaling Molecules

The addition of QS signaling molecules is the most well-known method for QS enhancement in WWTs that has direct control over QS levels and helps improve the bioreactor’s performance [58]. For instance, in a study, nitrogen transformation was successfully enhanced in an anaerobic ammonium oxidation process and moving bed biofilm reactor (MBBR) through the addition of exogenous C6-HSL and C8-HSL [129,130]. The addition of QS signaling molecules significantly improved the electrochemical reactor’s performance in organics removal. In microbial electrolysis cells (MECs) and microbial fuel cells (MFCs), power production capacity and electron transfer were found to have a stable increase with the addition of QS signaling molecules [62,113,131]. Disadvantages with this method are the high expense and instability caused by QQ bacteria [132].

#### 5.1.2. Adding Accelerators for Synthesis of QS Signaling Molecules

The addition of accelerators is another method for QS enhancement in WWTs, where they help increase the synthesis of QS signaling molecules. Advantages of using this method are their low-cost, compared to the method of direct addition of QS signaling molecules, and their biggest flaw is their failure in the synthesis of QS signaling molecules, which is a complex process and may not be productive. Some of the reported accelerators either act as precursors for QS molecules or work in their release. For example, a prominent QS enhancer is boron, which forms a complex with 4,5-dihydroxy-2,3-pentanedione (DPD), acting as a precursor for AI-2 activation [133,134]. In bioelectrochemical fuel cells, this addition of boron has resulted in an increased potential of almost 15 mV [134].

#### 5.1.3. QS Bacterial Cultivation

Another method that is more economical compared to direct addition of QS molecules or accelerators is the cultivation of QS bacteria, which are found in nature. In a study on leachates, Soler et al. (2018) isolated five QS bacterial strains out of a total of 99 bacterial strains [132]. While working on mature aerobic granular sludge, Zhang et al. (2020) used a supernatant of seven AHL-producing strains and added it to sequencing batch reactors [135]. He found an increase of 23%, 81%, and 27%, respectively, in concentrations of C6-HSL, C8-HSL, and N-(3- oxooctanoyl)-l-homoserine lactone (3OC8-HSL). Yong and Zhong, 2010 added *P. aeruginosa* (serves as pollutant degrader and AHL producer) to WWTs and found that it had a significant influence on the removal of organic carbon, nitrogen, and ammonia in the system and improved the overall performance of biological treatment systems [81]. The disadvantage of using this method is that these bacteria are more difficult to work with than other QS enhancement methods, may be washed out with excess biomass, and may be inhibited by competition with other bacteria in WWTs. In MBBR, the addition of *Sphingomonas rubra* sp. Nov (QS bacterium) had no significant effect on NH4^+^-N removal and COD. A similar scenario was also observed with the addition of *Aeromonas* sp. (an AHL producer), which caused a significant decrease in COD removal from 7% to 1%, from day 7 to day 40 [136,137].

### 5.2. QS Inhibition Methods

These methods aim to lower the concentration of QS signaling molecules. They either degrade QS signaling molecules, interfere with their functions, or inhibit their synthesis in biological treatment systems (Figure 4) [117]. QS inhibition can be achieved in four ways: (i) QS inhibitor addition, (ii) QQ bacteria cultivation, (iii) reactive oxygen species-based QS signaling molecules degradation, iv) enzyme-based QS signaling molecules degradation.

#### 5.2.1. QS Inhibitor Addition

These inhibitors interfere with QS receptors or inactivate QS signaling molecules and are thus being used in WWTs [138,139,140]. Examples of such inhibitors are homoserine lactone-like TGK series, cladodionen, 3-amino2-oxazolidinone YXL-13, ε-polysine, gingerol, aporphinoid alkaloids, etc. [141,142,143,144,145,146,147,148,149,150]. A QS inhibitor, 4- hydroxy-3-methoxy benzaldehyde (Vanillin), when applied to reverse osmosis (RO) membrane, decreases biofilm formation by up to 45%. In MBR, the addition of 100 ug/L of QS inhibitor 3,3′,4′,5-tetrachlorosalicylanilide caused a 50% decline in biofilm formation and a 30% decrease in AI-2 concentration. They are easily synthesized and economical, with cheap operating costs [117].

#### 5.2.2. QQ Bacterial Cultivation

A more common method than QS inhibitor addition is QQ bacterial strain cultivation for successful degradation of QS signaling molecules. Some examples of naturally isolated QQ bacteria that can rapidly degrade QS signaling molecules are *Penicillium restrictum* CBS 367.48, *Rhodococcus* sp. BH4, *Pseudomonas* sp. HS-18, and *Bacillus licheniformis* T-1 [97,151,152,153,154]. Among them, one of the most well-known QQ strains is *Rhodococcus* sp. BH4 [97,155]. Genetic engineering has also been employed in producing new and more potent QQ strains through plasmid transformation [156]. Just in a span of 30–60 s, the suspensions of Firmicutes and Betaproteobacteria at OD600 = 1.0 removed around 200 nM AHL [157]. It is expected that with further research on areas related to the isolation and evaluation of these QQ strains, more promising QS inhibition methods can be developed to be applied on a large scale [117,157].

#### 5.2.3. Degrading QS Signaling Molecules by Production of Reactive Oxygen Species (ROS)

In recent years, reactive oxygen species (ROS) production has gained popularity as a new QS inhibition method. ROS mentioned here are hydroxyl radicals and superoxide [158,159]. In *E. coli*, Short-time UV-TiO2 photocatalysis was used to generate ROS, which then not only inactivated A1-2 produced by *E. coli* but also caused a reduction of 42.6% in biofilm biomass [160]. In MBR, biofouling was successfully mitigated by continuous application of UV photolysis or photocatalysis [159,161].

#### 5.2.4. Degrading QS Signaling Molecules by Enzymes

The direct addition of enzymes for QS signaling molecule degradation is another new, emerging QS inhibition method. Many such enzymes have been studied so far; among them, enzymes for AHL degradation are the most studied. Acylase, deaminase, Lactonase, and decarboxylase are found to have the most capacity for AHL degradation in WWTs (Table 4) [103]. In MBR, levels of QS signaling molecules were reduced when Acylase (the most frequently used enzyme for AHL degradation) was applied [162].

## 6. Potential Applications of QS Regulated Biofilms for WWTs

### 6.1. Membrane Bioreactors (MBRs) 

Membrane bioreactors (MBRs) have been widely employed in drinking water production, wastewater reclamation, and seawater desalination, due to their compact design process, and the high water quality they produce [168]. However, one of the most significant operational issues associated with MBRs is membrane biofouling, which sometimes causes permeate flux and a decrease in water quality. It may also increase operational expenses to meet the need for periodic membrane cleaning and replacement [169,170,171,172]. Membrane biofouling refers to the unwanted buildup of microorganisms on membrane surfaces as a result of the adherence, growth, and proliferation of living organisms [171,173]. Biofouling is rarely avoidable, because most of the time systems are not sterile, allowing microorganisms to reseed and regrow at the expense of biodegradable substances present in water, turning them into metabolic products and biomass. Hence, biofouling is more difficult to remove, as compared to the fouling of organic and inorganic substances, which can be removed in the majority of cases by effective pretreatment. Therefore, efficient methods of biofouling control in the membrane process are required [4]. Studies have demonstrated that the issue of membrane biofouling in MBRs is highly associated with the production of AHL signals during biofilm formation [75,174,175,176,177,178]. In a study by Yeon et al. (2009a), TLC chromatographic analysis of the MBR membrane biocake showed the existence of C6-HSL and C8-HSL autoinducers produced during the operation of MBRs and increased the transmembrane pressure [106]. The ability of several bacterial species found in MBRs, including *Citrobacter*, *Enterobacter*, *Raoultella*, *Serratia*, *Pseudomonas*, *Aeromonas,* and *Klebsiella*, to create AHL signals and control the development of biofilms on membrane surfaces was also documented [69]. According to those researches, AHL-based QS uniformly controls the development of biofilms, which then causes membrane biofouling in MBRs [94].

One of the best techniques for preventing membrane befouling is quorum quenching (QQ), a technique of blocking the QS system and inhibiting gene expression (Figure 5) [172]. 

According to quorum quenching principles, quorum sensing interference in bacteria prevents the establishment of desired phenotypes such as biofilms [178]. QQ solves the limitations of high-cost operation, less resource utilization, and inconvenient management of traditional antifouling methods such as membrane cleaning, membrane modification, tuning of liquor, etc. [172,179,180,181,182,183,184]. In addition, it also eliminates the use of antimicrobial agents, hence reducing the risk of multi-drug resistance development in the biofilms [178,185,186]. Researchers have identified several AHL-generating bacteria and isolated them to be employed in biofilm reactors for wastewater treatment. Li et al., 2007 isolated two AHL-producing bacterial strains when they used the biofilm method for the treatment of nitrobenzoic acid-containing wastewater [187]. In order to screen out 200 microorganisms that can produce AHLs, Lade et al., 2014 employed the reported strains *Chromobacterium violaceum* CV026 and *Agrobacterium tumefaciens* A136 in WWT [69]. The results obtained after performing 16S rRNA sequencing allowed the speculation that *Aeromonas* and *Enterobacter sp*. were the dominating microorganisms in the system. The presence of AHLs indicates the existence of QS in the biofilm process under natural conditions [88].

#### Hybrid MBR-Attached Growth MBR (AGMBR)

Although MBR is popular now, some obstacles have interfered with conventional MBR (CMBR) commercialization for wastewater treatment. One major obstacle is membrane fouling; some other issues are the limited and low removal of some emerging pollutants [188,189] and significant removal of nitrogen and phosphorus from treatment systems [190]. Researchers have developed a hybrid MBR named attached growth MBR (AGMBR), through the integration of CMBR with an attached growth process, to overcome these limitations of CMBRs [188]. In this hybrid MBR, pollutant removal is accomplished through several ways such as (i) activated sludge-mediated sorption or degradation, (ii) degradation through microbes attached to media, (iii) by sorption onto the media, (iv) degradation through attached biofilms, and (v) rejection by a membrane. It was found that the use of media in AGMBR can improve the overall performance of the system in terms of membrane filterability and mitigating membrane fouling through modifying sludge suspension characteristics, reducing the concentration of suspended solids, and enhancing the physical scouring effect of the suspended carriers [191,192,193]. AGMBR mitigated membrane fouling through bio-augmented carriers such as PVA gel beads, PVC carriers, and MPCs. All of these carriers either control or reduce membrane fouling by allowing less organic matter buildup on the membrane surface, which in turn limits microbial growth on surfaces and reduces EPS and SMP levels [194,195,196,197].

In AGMBR, activated carbon (AC) and polymeric carriers were used as conventional media [188,198,199]. It was found that with the use of AC, AGMBR (both aerobic and anaerobic) can remove >80–90% of COD and BOD5 from municipal or domestic wastewater, in comparison to single aerobic MBR or AnMBR. The overall use of conventional media allows AGMBR to have increased biomass retention, diverse microbial communities, and improved microbial activity, all of which help in increasing the removal of pollutants [193]. New media used in AGMBR include biochar and bio-augmented carriers (containing selected strains/mixed cultures). These aid in thick biofilm formation and in removing high nitrogen content and COD (>80%). It was observed that biochar addition in anaerobic AGMBR favors removal of NH4^+^-N through adsorption, while in aerobic AGMBR, it helps remove up to 100% NH4^+^-N [200,201]. Hybrid MBR with biochar has the ability to not only remove conventional pollutants, but also emerging pollutants that are highly toxic and can persist in water bodies for long periods. Examples of such pollutants are adsorbable organic halogens, micropollutants, and nonylphenol. Wastewater also has some other compounds (refractory compounds) that are carcinogenic and mutagenic in nature, such as long-chain hydrocarbons, nitrogenous heterocyclic compounds, polynuclear aromatic hydrocarbons, etc. All of these pollutants have a detrimental effect on human health and are a threat to the environment [202,203,204]. In AGMBR, in comparison to CMBR, better and far more improved removal of organics, nutrients, and micropollutants has been reported due to an increase in biomass concentration, improved microbial activity and/or diversity, and improved simultaneous nitrification and denitrification [205,206]. Though AGMBR has widespread applications in controlling membrane fouling and wastewater treatment, there is still much work to do, such as researching new media for better applications of AGMBR for pollutant removal, both conventional and emerging, or for pilot or full-scale applications [193].

### 6.2. Microbial Fuel Cells (MFCs)

These are bio-electrochemical systems that use biofilms as catalysts to convert chemical energy into electrical energy. They can generate electricity through low-grade biomass and even through wastewater [207]. They have a cathode chamber, an anaerobic anode chamber, a salt bridge, or a proton exchange membrane (PEM) that works as a separating unit between both chambers and only allow protons (H^+^) to pass to cathode from anode. In MFCs, the anode is the final electron acceptor, and bacteria get their energy through electron transfer to the anode from their central metabolic system. With the help of an external circuit, the electrons are conducted to the cathode, where they form water after combining H^+^ and oxygen. Not only pure but mixed bacterial cultures are also being employed for electricity generation in MFCs [208,209,210,211,212]. There are three ways by which extracellular electron transfer from bacteria to an anode can be accomplished in an MFC; (i) with the help of pili (electrically conductive), (ii) with the use of electron mediators (provided artificially or produced by microbes), and (iii) through direct outer membrane c-type mediated cytochrome transfer [5,213,214]. In the case of two-chamber conventional MFCs, in the anaerobic anode chamber, biofilms oxidize their organic or inorganic substrates, and electrons released during this process are sent to the electrode directly or through an electron mediator indirectly. Electricity is generated when electrical current flows in from an electrical circuit [215].

In *P. aeruginosa*, phenazines act as electron mediators and promote respiration with the electrode. In MFCs, current production by *P. aeruginosa* and QS are directly linked with the production of phenazine [216,217]. Other than controlling rhlI, LasR, a transcriptional regulator, regulates the pqsABCDE operon positively to produce PQS, another QS signal for *P. aeruginosa*. In contrast to this, phz, which controls the production of phenazine and helps in electron transfer to the anode, is controlled by pqs operon [207]. In a study, for QS overexpression, multi-copy broad-host plasmid pYC-rhlIR was transformed in strain CGMCC 1.860. The transformation resulted in phenazine overproduction and elimination of the above-mentioned compound. This newly transformed strain is 1.6 times more potent than the wild-type strain in producing current output. A limiting factor in *P. aeruginosa* mediated electricity production is that PQS-driven QS can inhibit anaerobic growth of not only *P. aeruginosa*, but some other co-culture species [218,219]. A new *P. aeruginosa* strain with defective PQS and overproduction of phenazine was constructed to solve this problem. It can not only anaerobically synthesize higher phenazine concentrations, but also generate five times more current density in comparison to its parent strain [215]. Further electrochemical studies showed a correlation between phenazines’ overproduction and current increase. Correspondingly, microbial electrochemical systems (MESs) were shown to be affected by AHLs. In microbial electrolysis cells (MECs), AHL addition regulated biofilm formation at anodes, which enhanced bioelectrochemical activities in MECs [113].

For example, addition of a short-chain AHL, i.e., 3-oxo-C6-HSL, in comparison to the addition of 3-oxo-C12-HSL, provided higher hydrogen yields, which were due to changes in the community structure of microbes in cathodic biofilms [220]. Short-chain ‘3-oxo-C6-HSL’ AHL addition reduces hydrogen scavengers and increases the electrochemically active population of bacteria, which in turn give rise to higher hydrogen yield and electron recovery. In another study, by inoculating *P. aeruginosa* PAO1 pure culture in MFCs, it was found that compared to wild type, AHL-deficient lasIrhlI mutant type showed less extracellular electron transfer. Interestingly, electricity production was restored to the levels of wild-type strains with exogenous C4-HSL addition [221]. 

Wastewater contains sufficient amounts of organic compounds that can be used as substrates by MFCs for electricity generation. Wastewater of many different types has been used in MFCs for side-by-side water treatment end electricity generation. Rather than removing waste from wastewater through chemical or physical means, MFCs are an alternative, environmentally friendly way to harness the power of microbes for wastewater treatment and production of electricity. Substrates used for pollutant removal and energy generation by MFCs include starch and food processing wastewater, chocolate industry wastewater, domestic wastewater, azo dyes containing textile wastewater, and mustard tuber wastewater. However, there are still some limitations associated with full-scale use of MFCs, such as low power density, membrane fouling, and high internal resistance of the reactor. These limitations have limited MFC application on a commercial scale [222]. 

### 6.3. Granular Sludge

Granular sludge is a particular type of biofilm that can be categorized into aerobic granular sludge (AGS) and anaerobic granular sludge (AnGS). All of the previous studies on granular sludge primarily studied the effect of fluctuating environmental conditions and physicochemical properties of sludge on microbial communities, hydrophobicity, EPS, surface charges, and the granulation process. However, in the sludge granulation process, regulation of QS on biofilms is also applied. The primary benefit of QS in flocculent granular sludge includes the increased production of EPS and ATP, which ultimately speed up the formation of granular sludge microbial communities. In granular sludge, the microbial communities consists of several species that perform a variety of metabolic tasks demonstrating the community level as a whole. Signaling molecules are the vital link that binds these communities together. Currently, the major signaling molecules in granular sludge systems are diffusible signaling molecules (DSF), intraspecific signaling molecules (AHL), and interspecific signaling molecules (AI-2) [88].

#### 6.3.1. Aerobic Granular Sludge (AGS)

Aerobic granular sludge has gained extraordinary attention in the research field because of its superior settling ability, high biomass retention, outstanding anti-shock loading capability, small occupied footprint, and treatment efficiency as compared to conventional activated sludge. To cultivate aerobic granular sludge, activated sludge is used as a seed. AGS grows as a spherical form of biofilm with no carriers because of its self-immobilization property [88,223]. The pilot and full-scale use of AGS persuasively demonstrate the value of aerobic granulation technology in real-world settings [224]. Several potential factors, including organic loading rates, substrate composition, hydrodynamic shear force, settling time, seed sludge properties [225,226], starvation period, and sludge discharge affect aerobic granulation and have been studied [227,228,229]. During aerobic granulation, selection pressures are indispensable to achieve fast-settling aerobic granules while washing out the slow-settling flocs [229]. Moreover, the bacterial interactions and their coordinated behaviors also affect the aerobic granulation setting [230].

In 2006, for the first time, a report was published related to aerobic granulation and QS. In that report, AI-2 was detected in two genetically distinct bacterial co-culture strains forming aerobic granules [231]. Besides AI-2, AHLs were also detected in the cellular extracts and suspensions of AGS [230]. Extensive data are available online explaining the characteristics and different cultivation methods of AGS. However, the involved mechanism is not yet clear. The evidence for the vital role of QS in granule formation is increasing day by day. AHL displays a dense signaling molecule network within the granule interior and performs well in boosting the development of biofilms and microbial granulation [59]. Signaling molecules continue to promote the adhesion development and accumulation of suspended bacteria to form mature granules. They also aid in preserving the stability of granule structure and speed up granule formation. More signaling molecules are detected in mature granules as compared to smaller granules [88]. Hence, QS-based regulation may prove to be a novel technique to encourage AGS development and preserve granular stability, but there are still some challenges and knowledge gaps that need to be addressed.

#### 6.3.2. Anaerobic Granular Sludge (AnGS)

Anaerobic granular sludge has a more complicated quorum sensing system than AGS and still lacks a relatively full regulation mechanism. AHL, AI-2, and DSF are the main research topics associated with the AnGS system. Ding et al. [231], reported 5.72 ± 1.56 pM/L as the initial total concentration of AHLs produced in a granular sludge system, of which 95% of the AHL concentration was composed of C4-HSL [231]. However, the C4-HSL concentration dramatically decreased with the development of AnGS, while the content of AI-2 remained stable [232,233,234]. Another study reported an increase in DSF concentration to 1.76 ± 0.18 nM/L from 0.66 ± 0.06 nM/L [235].

In AnGS, the granule particle size is influenced by AHL-based regulation. A large-sized particle has high biological activity, and its internal structure characterization based on its small diffusion area, porosity, low diffusivity, and macro void, facilitates improved production of biogas [236]. In contrast to this, the inoculation of exogenous AHL-producing and -quenching bacteria reduced the particle size in a study that hinted toward further investigation in this regard. This may have resulted due to the destruction of the original bacterial community structure as the additional strains did not dominate the system and showed less competition with local bacteria. Ding et al. [231] reported an increase in granular diameter via AI-2 regulation while reduced granule size due to DSF regulation [231].

Under neutral/weak alkaline conditions, by increasing the amount of AI-2 and reducing the amount of DSF in the system, the relative hydrophobicity and strength of the granular sludge will increase. This also facilitates granular sludge formation with large-size granules. However, if there is an imbalanced nitrogen supply, increasing the amount of AI-2 and reducing the amount of DSF in the system significantly reduces the particle strength and hydrophobicity of the sludge [231,237,238]. The findings of Ding’s study [231,237,238] further supported the notion that AI-2 can encourage the synthesis of EPS and increase the particle diameter of granular sludge. Additionally, AHLs might encourage Methanothrix development in the upflow anaerobic sludge bed reactor (UASB), which would greatly enhance sludge granulation and reactor operating efficiency. Exogenous AHLs play a significant role in regulation of EPS and microbial community structure, improving the performance of AnGS. Specific AHLs can increase the organic matter removal capacity and methanation ability in AnGS [239,240].

## 7. Conclusions and Future Perspectives

Biofilms are the aggregations of the same or different microbial communities, impregnated in a complex matrix and partitioned by a network of water channels. These are natural habitats in which microbes interact with each other by exchanging genetic material, metabolites and signaling molecules. Biofilm formation is a complex process regulated by QS. The control of biofilm formation has gained much attention in WWT. In order to improve and optimize biofilm control strategies, it is important to understand the factors that strongly influence biofilm formation. Genetic engineering techniques, such as metabolic engineering, omic based approaches, genome editing and bioinformatics approaches, have opened new avenues for biofilm-related wastewater treatment research. QS regulates EPS and biosurfactant synthesis, which can significantly contribute in waste water treatment. Understanding the complex QS mechanisms is difficult, because of the production of multiple signaling molecules. Hence, the determination of the factors that stimulate the emergence of QS is a major challenge. These factors may include gene expression patterns, cell-to-cell interaction, physiological properties and molecular level details of signaling molecules. 

Although the utilization of QS for biological wastewater treatment has made great achievements in the field of WWT, there are still several problems yet to be addressed. Future research on biofilm-based wastewater treatment needs to cover several dimensions such as the evaluation of the risks associated with QS regulation, accurate estimation of QS level so that the interference caused by the coexistence of QQ and QS can be eliminated, evaluation of the efficacy of several QS inhibition or activation methods usually adopted in wastewater treatment, understanding the metabolism, distribution and fate of the QS signaling molecules produced during wastewater treatment, and changes in microbial community at the molecular level. The disturbance created by QQ in the understanding of QS should be eliminated by comprehensive and quantitative optimization for different types of wastewater bioreactors. The molecular tools, such as metagenomics and meta-transcriptomics, should be comprehensively adopted to understand the molecular level changes in QS-regulated biofilm during wastewater treatment. Most of the QS regulation-based studies of WWTs are not being conducted at a large scale, but at experimental levels in laboratories. Therefore, the mechanisms and effects of QS regulation strategies should also be explored on a large scale. Moreover, the QS approach, besides having several advantages, must have some negative impacts on the physiological behavior and structure of the biofilm. These impacts should also be studied for improvements in biological wastewater treatment by eco-friendly approaches using QS-regulated biofilms.

## Figures and Tables

**Figure 1 ijms-23-09751-f001:**
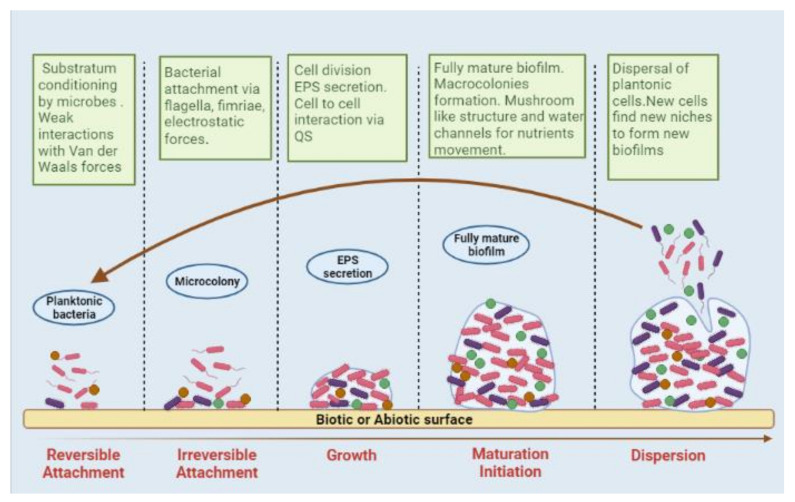
Schematic diagram of various steps involved in the biofilm formation process.

**Figure 2 ijms-23-09751-f002:**
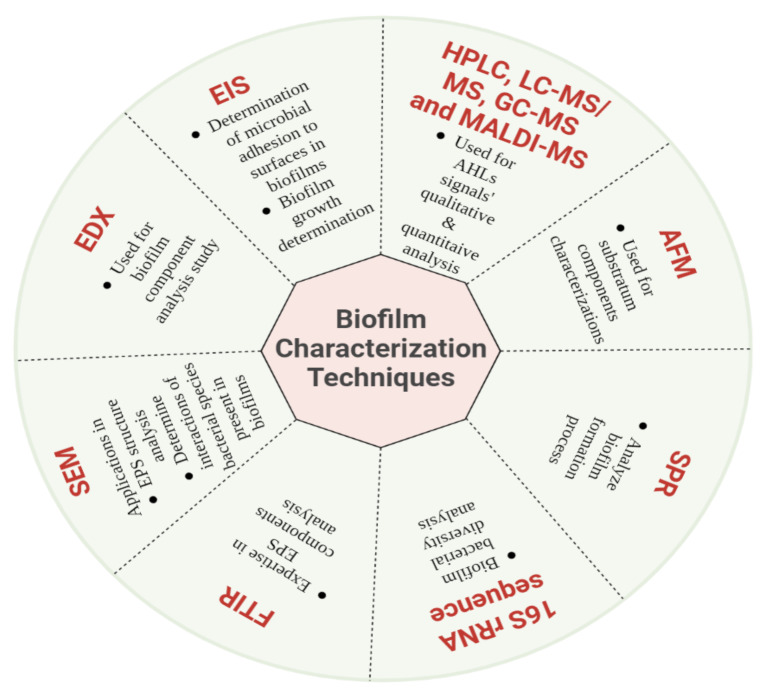
Schematic diagram of various techniques used for biofilm characterization.

**Figure 3 ijms-23-09751-f003:**
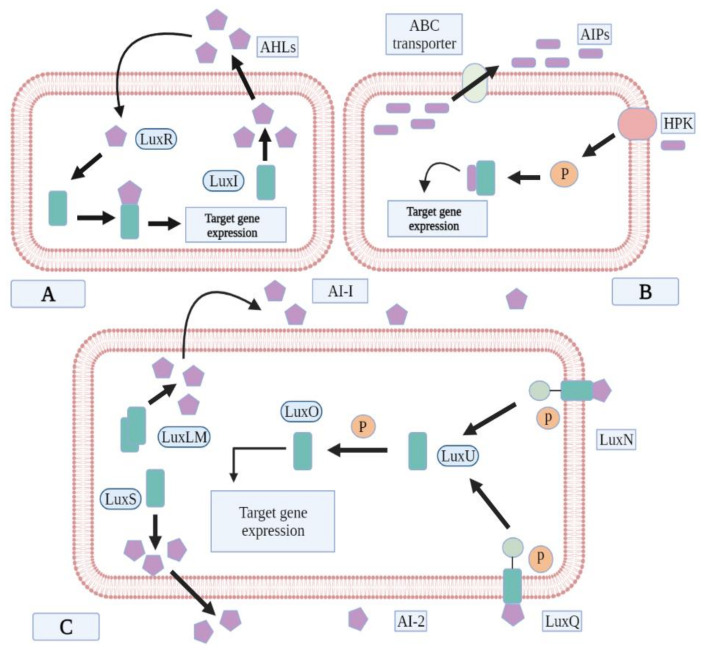
Typical QS signaling pathways. (**A**) AHL pathway seen in Gram-negative bacteria, (**B**) AIP pathway seen in Gram-positive bacteria, (**C**) AI-2 pathway seen in both Gram-negative and Gram-positive bacteria.

**Figure 4 ijms-23-09751-f004:**
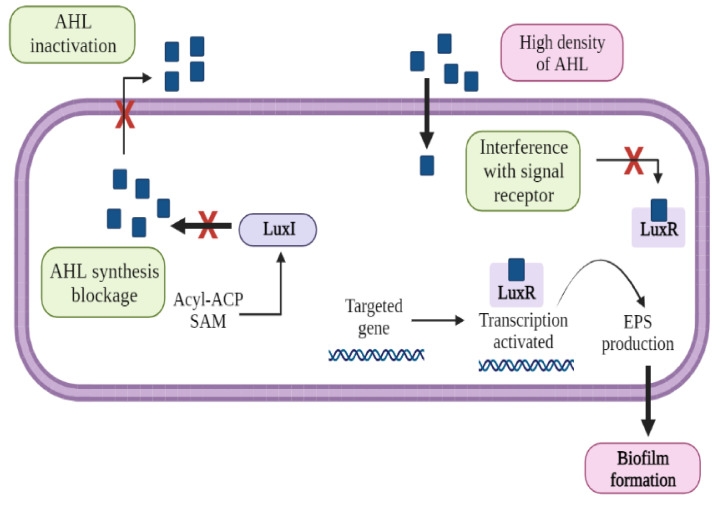
AHL degradation pathways. (1) AHL synthesis blockage, (2) AHL inactivation, (3) interference with AHL signal receptor.

**Figure 5 ijms-23-09751-f005:**
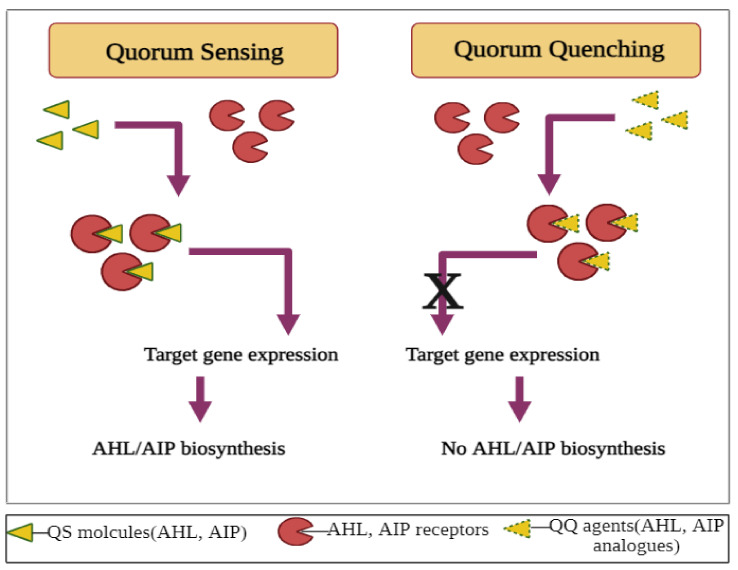
Diagrammatic representation of the difference between quorum sensing and quorum quenching.

**Table 1 ijms-23-09751-t001:** QS and QQ bacteria involved in wastewater treatment.

Type	Bacteria	Signal Type	QS/QQ-Related Activities	Treatment Processes	Wastewater Type	Ref.
Quorum sensing bacteria	*Sphingomonas paucimobilis*	-	-	EBPR aerobic-anaerobic process/conventional aerobic activated sludge process	Municipal or industry wastewater	[83,84]
	*Burkholderia* sp. DW2–1	CepI/CepR	Biosurfactant production	-	Municipal or industry wastewater	[85,86]
	*B. cenocepacia* BSP3					
	*Aeromonas*	AHL, AI-2	Biofilm formation	-	-	[54]
	*A. hydrophila*	C4-HSL, C6-HSL, AI-2	Biofilm development	Activated sludge process	Municipal wastewater	[66,76]
	*Pseudomona aeruginosa* PAO	N-decanoyl-L-HSL (C10HSL)	Anthranilate biodegradation in waste water	-	-	[87]
	*Xanthomonas* sp.	DSF	EPS production	Activated sludge	-	[88]
	*P. aeruginosa*	RhlI/RhlR	Degradation of phenol	-	Industrial and municipal wastewater	[22]
	*Vibrio* sp.	AI-2, AHK	Biofilm formation, virulence factor production	-	-	[54]
	*Ac. baumannii* strain M2	3- hydroxy C12-HSL	Biofilm formation and development, surface motility	-	Municipal or industry wastewater	[83,89,90,91]
	*Pseudomonas* sp.	C4-HSL, C6-HSL, oxo-C12-HSL, PQS	Biofilm formation and development; virulence factor production; EPS production; interspecies competition; denitrification	Activated sludge	-	[83,88]
	*Nitrosomonas europaea*	C6-HSL, C8-HSL C10-HSL	-	Activated sludge process	Industrial wastewater	[92,93]
	*P. putida*	C4-HSL,	Biofilm formation	Activated sludge process	Municipal wastewater	[94]
Quorum quenching bacteria	*Variovorax paradoxus* strain VAI-C	C4-HSL, C6-HSL C8-HSL, C10-HSL C12-HSL 3-oxo-C6-HSL	Degrade other species AHLs	Activated sludge process	Industrial wastewater	[92,95,96]
	*Rhodococcus* sp. BH4	C6-HSL, C8-HSL C10-HSL, C12-HSL 3-oxo-C6-HSL 3-oxo-C8-HSL 3-oxo-C10-HSL 3-oxo-C12-HSL	Inhibit biofilm formation in MBR	Real MBR plant	Municipal wastewater	[72,97,98,99,100]
	*Acinetobacter* sp. strain Ooi24	C10-HSL	-	Activated sludge process	Unknown	[94]
	*Pseudomonas* sp. 1A1	C6-HSL,C8- HSL C10-HSL, C12-HSL 3-oxo-C8- HSL 3-oxo-C10- HSL 3-oxo-C12- HSL	Inhibit biofilm formation in MBR	Lab-scale MBR/activated sludge process	Municipal wastewater	[71,101,102]

Not available = ‘‘-‘’.

**Table 2 ijms-23-09751-t002:** Techniques for autoinducer detection, identification and characterization in wastewater treatment systems.

Technique Employed		Applications	Advantages	Limitations	Ref.
Bacterial biosensors	*C. violacum* CV026	Detects AHLs by producing purple colored pigment violacein	Simple bioassay	Unable to detect any of the three hydroxyl derivatives, no information on AHLs’ structure and concentrations	[103]
	*A. tumefaciens* A136 & *A. tumefaciens* NTL4	Produces blue spots on TLC plates or Petri dishes upon detection of AHLs	A fast biosensor for AHL screening	Not capable of detecting N-butanoyl-homoserine lactone, no information on AHLs’ structure and concentrations	[94,103]
Luminescence		Beta-Glo Assay System (Promega, Madison, WI, USA) based AHL quantification technique	Simple, easy, and fast bioassay	NA	[72]
TLC		Partial characterization and structure identification of AHLs	Easy, fast, and cheap method for determining preliminary structure information, can be coupled with biosensor or sulfuric acid	TLC alone is enough to determine the complete structure of AHLs	[104,105,106]
HPLC		Can detect a large no. of AHLs	Simple and fast technique for AHLs’ qualitative and quantitative analysis	Unable to provide an AHL-specific structure.	[103]
HPLC–MS/MS		Used for quantification of various AHLs	Provides rapid AHL quantification	NA	[107]
LC–MS		Able to determine AHL structure, can be coupled with HPLC	Can quantify very small amounts of AHLs—down to picomoles	Some qualitative information and chromophores are needed for operation	[72,103]
SPE-LLE with LC-MS/MS		Detection and trace analysis of AHLs in wastewater	Can identify and quantify AHL trace levels in wastewater systems	NA	[108]
ELISA		Quantitative detection of AHLs and their degradation products	Rapid, cheap, and sensitive method, needs low amounts of sample (<1 mL)	NA	[109]
HLB and UE coupled with UPLC-MS/MS		AHL detection	Robust and sensitive method for AHL detection in wastewater	NA	[110]
UHPLC-MS		AHL detection and quantification	Independent of the sample matrix, can detect low concentrations of AHLs	NA	[109]
IR		Identifies functional groups	Simple, cheap, easy, and versatile	Sample preparation needs extra care	[40]
NMR		Used in AHL structure determination	Can detect fine details of structural components	Time-consuming and costly	[111]
GC-MS, NanoLC-MS/MS, MALDI-MS, and magnetic molecularly imprinted polymer nanoparticles based electrochemical sensor		AHL detection and characterization	Provides accurate details on AHL structure and characterization	NA	[94,112]

Legend: TLC—thin layer chromatography; HPLC—high-performance liquid chromatography; HPLC–MS/MS—LC-MS—liquid chromatography–mass spectrometry; SPE-LLE—solid phase extraction—liquid–liquid extraction; LC-MS/MS—liquid chromatography—tandem mass spectrometry; ELISA—enzyme-linked immunosorbent assay; HLB—*hydrophilic–lipophilic balanced sorbent*; UE—ultrasonic extraction; UHPLC-MS—ultrahigh-performance liquid chromatography coupled to mass spectrometry; IR—infrared; NMR—nuclear magnetic resonance; GC-MS—gas chromatography-mass spectrometry; NanoLC-MS/MS—nanoscale liquid chromatography coupled to tandem mass spectrometry; MALDI-MS—matrix assisted laser desorption/ionization.

**Table 4 ijms-23-09751-t004:** Major naturally existing quorum quenching enzymes used in wastewater treatment.

QQ Enzyme	Source Organism	Mechanism of Action	Ref.
AHL-acylase	*Tenacibaculum* discolor strain 20 J,	AHL degradation	[163]
	*Hyphamonas* sp. DG895	C4HSL and 3OC12-HSL	[103]
AHL-oxidase	*Bacillus megaterium*	C4HSL and 3OC12HSL	[164]
AI-2 kinase (LsrK)	*Escherichia coli*, other enteric bacteria	Degradation of AI-2	[112,165]
Lactones	*Streptomycetes* spp.	Mimics AHL signals	[166]
AHL-oxidoreductase	*Burkholderia* strain GG4	3OC6HSL	[167]
AHL-lactonase	*Halomonas* sp. strain 33	AHL degradation	[163]
	*Bacillus cereus*	AHL degradation	[103]

## Data Availability

Not applicable.

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
