# Peer review of "Exploring the Function of Quorum Sensing Regulated Biofilms in Biological Wastewater Treatment: A Review"

_ijms, 2022, doi:10.3390/ijms23179751_

Round 1

Reviewer 1 Report

The authors provide a comprehensive review on Quorum Sensing (QS) based bacterial contact independent cell-cell communication and their application to wastewater treatment. In this review, the author discussed the complete picture of the biofilm formation, and development under the impact of QS (exogenous molecules, accelerants, and culturing microbes) and Quorum quenching (QQ) bacteria (degrading enzymes, oxidants, and inhibitors) related to the WWT. I believe this review article can bridge the gaps in current investigations and their necessity for future endeavors.

Minor comments:

Page 3, lines # 64, 67 & 68 – discuss the stressful and or adverse environmental conditions. I would suggest they rephrase those into one sentence.

Page 7, line # 212-214: authors report the advancement in molecular biology and do not explicitly provide the details of the methodology advancement. It would be helpful for a broader audience to provide a few lines on advancement.

Page 7, line # 214: the word discovered is inappropriate; maybe change it to “observed”

Author Response

Reviewer #1

The authors provide a comprehensive review on Quorum Sensing (QS) based bacterial contact independent cell-cell communication and their application to wastewater treatment. In this review, the author discussed the complete picture of the biofilm formation, and development under the impact of QS (exogenous molecules, accelerants, and culturing microbes) and Quorum quenching (QQ) bacteria (degrading enzymes, oxidants, and inhibitors) related to the WWT. I believe this review article can bridge the gaps in current investigations and their necessity for future endeavors.

Our sincere thanks for taking the time to review this manuscript, and your close attention to detail. We highly appreciate your overall positive feed-back regarding the quality of the manuscript! Please see below for our responses to your comments:

Minor comments:

Page 3, lines # 64, 67 & 68 – discuss the stressful and or adverse environmental conditions. I would suggest they rephrase those into one sentence.

According to reviewer suggestion the sentences was rephrased into one sentence (see the lines 67-69 of the revised version).

Page 7, line # 212-214: authors report the advancement in molecular biology and do not explicitly provide the details of the methodology advancement. It would be helpful for a broader audience to provide a few lines on advancement.

The authors completely agree the reviewer opinion. In order to address the raised concern, a completely new table (Table 2) was elaborated.

Page 7, line # 214: the word discovered is inappropriate; maybe change it to “observed”

Changed according to the reviewer suggestion.

We hope that all the concerns of the worthy reviewers have been addressed however, we are ready to further revise/modify the article if still exists some shortcomings.

Thank you again!

Reviewer 2 Report

The paper looks solid and well organized. I suppose it fits the scope of IJMS journal, so the paper may be considered for publication after revision in accordance to the comments below:

1. The paper supposedly does not fully correspond to the journal template.

2. There are some language corrections needed throughout the text.

3. The authors suggest MBR as a potential application of QS Regulated Biofilms. I think the paper will look more complete if to extend it by considering other treatment technologies.

Author Response

Reviewer #2

The paper looks solid and well organized. I suppose it fits the scope of IJMS journal, so the paper may be considered for publication after revision in accordance to the comments below:

Our sincere thanks for taking the time to review this manuscript, and your close attention to detail. We highly appreciate your overall positive feed-back regarding the quality of the manuscript! Please see below for our responses to your comments:

  1. The paper supposedly does not fully correspond to the journal template.

During the manuscript preparation, the authors used the indicated Microsoft Word Template by the IJMS journal (https://www.mdpi.com/journal/ijms/instructions), and according to their knowledge/opinion, there are no major deviations from guideline. If yes, please indicate the necessary improvements. Likewise, the possible nonconformities, are solvable during „Final Proofreading Before Publication” status.

  1. There are some language corrections needed throughout the text.

The authors acknowledge the fact that the English content of the manuscript need some improvements. Accordingly, during the manuscript revision, each member of the research team tried to do her/his best to improve the English language. In addition, in the lack of “extensive English edits” raised concern by the reviewer, the authors believe that the improvement of the English language is solvable during the final “Pending English” status before publication, when the paper will be edited and finalized by the MDPI team. Thank you for your understanding and consideration!

  1. The authors suggest MBR as a potential application of QS Regulated Biofilms. I think the paper will look more complete if to extend it by considering other treatment technologies.

The authors agree the reviewer suggestion! Therefore, the raised concerns were fully addressed. For this, please see the newly elaborated sections 6.1.2 and 6.1.3. (the lines 475-610 in the revised version). In addition, another 36 new references were inserted in the reference list.

We hope that all the concerns of the worthy reviewers have been addressed however, we are ready to further revise/modify the article if still exists some shortcomings.

Thank you again!
